# Non-indicated vitamin B$_{12}$- and D-testing among Dutch hospital clinicians: a cross-sectional analysis in data registries

Joris LJM Müskens [1], Rudolf Bertijn Kool [1], Gert P Westert,[1] Maarten Zaal,[2] Hein Muller,[2] Femke Atsma [1], S A van Dulmen [1]

¹IQ Health science department, Radboud University Medical Centre, Nijmegen, The Netherlands
²Dutch Hospital Data, Utrecht, The Netherlands

**Correspondence to**
Joris LJM Müskens;
Joris.muskens@radboudumc.nl

## ABSTRACT

**Objectives** To assess the extent of non-indicated vitamin B$_{12}$- and D-testing among Dutch clinicians and its variation among hospitals.

**Design** Cross-sectional study using registration data from 2015 to 2019.

**Participants** Patients aged between 18 and 70 years who received a vitamin B$_{12}$- or D-test.

**Primary and secondary outcome measures** The proportion of non-indicated vitamin B$_{12}$- and D-testing among Dutch clinicians and its variation between hospitals (n=68) over 2015–2019.

**Results** Between 2015 and 2019, at least 79.0% of all vitamin B$_{12}$-tests and 82.0% of vitamin D-tests lacked a clear indication. The number of vitamin B$_{12}$-tests increased by 2.0% over the examined period, while the number of D-tests increased by 12.2%. The proportion of the unexplained variation in non-indicated vitamin B$_{12}$- and D-tests that can be ascribed to differences between hospitals remained low. Intraclass correlation coefficients ranged between 0.072 and 0.085 and 0.081 and 0.096 for non-indicated vitamin B$_{12}$- and D-tests, respectively. The included casemix variables patient age, gender, socioeconomic status and hospital size only accounted for a small part of the unexplained variation in non-indicated testing. Additionally, a significant correlation was observed in non-indicated vitamin B$_{12}$- and D-testing among the included hospitals.

**Conclusion** Hospital clinicians order vitamin B$_{12}$- and D-tests without a clear indication on a large scale. Only a small proportion of the unexplained variation could be attributed to differences between hospitals.

## INTRODUCTION

Low-value diagnostic testing is a serious problem in most healthcare systems.[1] Low-value care is defined as care that offers no net benefit for the patient and which can be associated with harmful outcomes and wasteful spending.[2–4] Within diagnostic testing, studies show that both vitamin B$_{12}$- and D-tests are frequently ordered within medical practice.[5–8] However, there are only a few indications that justify the ordering of a vitamin B$_{12}$- or

## STRENGTHS AND LIMITATIONS OF THIS STUDY

⇒ This study has examined the ordering and appropriateness of vitamin B$_{12}$- and D-testing among hospital clinicians and its variation using data over several consecutive years.

⇒ The use of a detailed nationally representative database allowed us to generate an accurate and reliable overview of the extent of (non-indicated) vitamin B$_{12}$- and D-testing among exclusively hospital clinicians within the Dutch healthcare system.

⇒ Additionally, the use of registration data derived directly from hospital registration systems enabled us to accurately distinguish appropriate from inappropriate vitamin testing using diagnosis codes rather than the proxy of testing interval.

D-test which are described in international guidelines. Among healthy adults, almost no indications for vitamin B$_{12}$- and D-testing exist.[9 10] International guidelines for hospital clinicians clearly state that vitamin B$_{12}$- and D-tests should only be ordered in specific scenarios, such as in patients with Coeliac or Crohn's disease, and not in patients with only vague complaints.[9 11–14] Assessments indicate that as much as 77% of vitamin B$_{12}$-tests and 91% of vitamin D-tests could lack an indication.[7 15–23]

Studies show that between 8% and 28% of vitamin B$_{12}$-tests and between 6% and 91% of vitamin D-tests could be non-indicated. Most of these studies have been conducted among general practitioners (GPs).[17 19 21 23] Only a few studies have clearly specified that their assessment of the volume of non-indicated vitamin testing concerned hospital clinicians alone. Two studies from Italy reported that on average 17% and 6% of vitamin B$_{12}$- and D-tests were potentially non-indicated.[7 18] However, these assessments used data from a single hospital and therefore do not provide a representative overview. Furthermore, these

studies based their assessment of appropriateness on a proxy such as testing interval and not directly by examining the diagnosis for which the test was requested.

In order to address non-indicated vitamin $B_{12}$- and D-testing, an accurate assessment of the magnitude and variation of non-indicated testing is needed. We therefore aimed to assess both the volume and proportions of patients aged between 18 and 70 years who potentially received non-indicated vitamin $B_{12}$- and D-tests among Dutch hospital clinicians between 2015 and 2019 using clinical registration data. We also examined the hospital variation in non-indicated vitamin $B_{12}$- and D-testing in order to identify opportunities for improvement. The proportion of outpatient visits receiving a non-indicated vitamin $B_{12}$- or D-test among the hospitals included, and the diagnosis codes often associated with vitamin $B_{12}$- and D-testing over 2019 were also explored. In doing so, we aimed to gain more insight into the volume of non-indicated vitamin testing among hospital clinicians and the diagnoses underlying such testing.

## METHODS
### Design and database
We conducted a cross-sectional study using registration data among Dutch hospital clinicians between 2015 and 2019. Data were obtained from the Dutch National Basic Hospital Care Registration (*Landelijke Basisregistratie Ziekenhuiszorg, LBZ*).[24] The LBZ contains medical, financial and administrative information from all patients undergoing treatment in any Dutch hospital. All vitamin $B_{12}$- and D-tests ordered by clinicians in Dutch hospitals over the examined time period were extracted; including the associated diagnosis codes, patient age, gender, socioeconomic status (SES) and hospital size. After consulting paediatricians and geriatricians, we limited our analysis to patients aged 18–70 years. Both paediatricians and geriatricians indicated that for patients below 18 and above 70 years, there are many regional screening protocols which also often vary between hospitals. Patients were assigned to one of three age categories; 18–29, 30–49 and 50–70 years. SES scores were derived from a table containing SES scores on the level of four-digit postal codes as published by the Dutch Institute for Social Research in 2017.[25] Patients were also assigned to one of six SES categories, based on quintiles calculated with the SES information from all Dutch neighbourhoods. Hospital size was operationalised by assigning a hospital to one of three categories (small/medium/large) based on tertiles calculated using the number of outpatient visits encountered over 2019.

### Patient and public involvement
No patients or members of the public were directly involved in the study. Owing to the nature of this study and data privacy constraints, no patients or members of the public were involved in the study design, analysis, interpretation of data or revision of the manuscript.

### Analysis of trends in proportions of non-indicated vitamin $B_{12}$- and D-testing among hospital clinicians
For the assessment of the proportion of justified indication versus non-indicated vitamin $B_{12}$- and D-testing, we used a service lens, as previously described by Chalmers *et al.*[26] This entails that all vitamin $B_{12}$- or D-tests ordered were included in our denominator and all vitamin $B_{12}$- and D-tests ordered with no indication in our numerator. For our distinction of non-indicated vitamin testing, we followed several steps. First, all recommendations regarding vitamin $B_{12}$- or D-testing were extracted from the relevant guidelines. Initially, we reviewed the Dutch guidelines of hospital clinicians published by the Federation of Medical Specialists for indications for vitamin $B_{12}$- and D-testing.[10] We managed to identify little to no recommendations concerning indications regarding the use of vitamin $B_{12}$- or D-tests. We therefore chose to supplement these with indications derived from Dutch GP guidelines.[13 14] Second, the ICD10 codes corresponding to the diagnoses which warrant the ordering of a vitamin $B_{12}$- or D-test were collected from these recommendations. Third, the resulting list of ICD10 codes was reviewed by the involved experts to prevent missing relevant codes or diagnoses. We consulted two expert clinicians (an internal medicine physician and a haematologist) in the process of generating the list of indications justifying a vitamin $B_{12}$- or D-test. Fourth, after completing the list of ICD10 codes, all Clinical Classification Software (CCS) codes associated with these ICD10 codes were extracted from the LBZ database. Subsequently, all Diagnosis-Treatment Combination (DTC) codes associated with the list of relevant CCS codes were extracted. The resulting list was, again, checked for completeness before starting with the assessment of indicated vitamin tests. This process was repeated until all researchers and clinicians agreed on the accuracy of the list of indications justifying vitamin $B_{12}$- and D-testing. Online supplemental file 1 lists ICD10 and DTC codes used to determine the appropriateness of the identified vitamin $B_{12}$- or D-tests. It also contains a description of how we identified and linked the identified ICD10 codes to each of the patients included. In an effort to provide potential handles for the design of interventions, we also examined which diagnosis codes were frequently associated with non-indicated tests.

### Assessment of hospital variation in non-indicated vitamin $B_{12}$- and D-testing
Hospital variation in non-indicated vitamin $B_{12}$- and D-testing was assessed using a multilevel logistic regression analysis, with a random effect for hospital. Separate models per year were made to assess whether the variation in non-indicated vitamin $B_{12}$- and D-testing was robust over time. Generalised variance inflation factors were calculated to test for collinearity among the included variables before multilevel analysis was conducted (online supplemental file 2). Models were adjusted for the casemix variables: patient age, gender, SES and hospital size. We corrected for patient age, gender and SES, as previous research

**Table 1** Overview of the study population characteristics of the used population over the entire period examined (2015–2019)

| General info regarding the used population between 2015 and 2019 | Number or proportion | Min | Max | Median | IQR |
|---|---|---|---|---|---|
| Total number of unique patients | 9 214 425 | N.A. | N.A. | N.A. | N.A. |
| Gender (% female) | 64.50% | N.A. | N.A. | N.A. | N.A. |
| Average number of patients among the hospitals (±SD) | 126 722 (±169 110) | 37 | 1 244 526 | 126 722 | 119 601.8 |
| Average number of unique patients among the hospitals (±SD) | 47 561 (±54 317.5) | 32 | 415 673 | 36 561 | 38 590 |
| Average age of the patients included (±SD) | 47.91 (±14.60) | 18 | 70 | N.A. | N.A. |
| Average SES category of the patients included (±SD) | 2.86 (±1.43) | 1 | 6 | N.A. | N.A. |
| Average no. of outpatient visits among the hospitals included | 1 721 337 (±819 148) | 470 117 | 4 659 172 | N.A. | N.A. |

SES, socioeconomic status.

showed that these affect the amount of care that patients require, receive and have access to.[27–31] We included a proxy for hospital size (eg, the total number of outpatient visits in each year) while recent evidence shows that larger healthcare providers tend to provide more low-value care.[32] Vitamin tests conducted in patients with a missing SES score or DTC code were excluded from the analysis. Intraclass correlation coefficients (ICCs) were calculated to assess which part of the unexplained variation in non-indicated vitamin $B_{12}$- and D-testing could be ascribed to differences between the included hospitals, using the method of Snijders and Bosker to assess the error variance.[33]

### Correlation in non-indicated vitamin testing over 2019

Additionally, we also examined whether a correlation existed between the proportions of outpatient visits that received a non-indicated vitamin $B_{12}$- or D-test over 2019 among the hospitals included. Correlations were assessed using the Pearson correlation coefficient for normally distributed variables and the Spearman correlation coefficient for non-normally distributed variables. Normality was assessed using both density plots and the Shapiro-Wilk test.

## RESULTS

### Volume vitamin tests among Dutch hospital clinicians

Table 1 provides a general overview of the population characteristics of the population included in our study. Between 2015 and 2019, the number of vitamin $B_{12}$- and D-tests ordered by clinicians increased by 2.0% (from 275 032 to 280 522) and 12.2% (from 300 013 to 336 736), respectively. A similar trend was also observed in the proportion of patients who received at least one vitamin $B_{12}$- or D-test, increasing by 2.5% and 11.3% over the examined period. The amount of vitamin $B_{12}$- and D-tests ordered among women remained almost twice as high compared with men over the entire period examined.

Table 2 provides an overview of the outcomes, and online supplemental file 3 contains a more detailed breakdown by gender, age and SES groups on the patient level. The number of patients with at least one vitamin test increases rapidly with age. The included patients, with at least one vitamin $B_{12}$- or D-test, showed to be more equally distributed over the SES categories. Only in the highest SES category, a slight decrease in the number of patients with a vitamin determination was observed.

### Non-indicated testing

Between 2015 and 2019, around 78% of the vitamin $B_{12}$-tests conducted among patients aged between 18 and 70 years, with a registered DTC code, lacked an indication. In the case of vitamin D-testing, around 82% of determinations had no clear indication. Although the number of vitamin tests is higher among women, no large differences in proportions of non-indicated testing were observed between genders. In the case of both age and SES, the proportion of patients with a non-indicated vitamin $B_{12}$- or D-test remains relatively constant across all groups over the study period (online supplemental file 3). With the proportion of non-indicated testing remaining around 80.0% for vitamin $B_{12}$ and 83.5% for vitamin D across the different age and SES. Our analysis of diagnosis codes that are most often associated with non-indicated vitamin $B_{12}$- and D-testing revealed that tests are ordered for various reasons. Similar diagnostic codes were associated with both non-indicated vitamin $B_{12}$- and D-tests, including general malaise, fatigue without diagnosis and ulcerative colitis. Online supplemental file 4 contains an overview of the top 20 diagnosis codes for both vitamin $B_{12}$- and D-tests.

### Hospital variation in non-indicated vitamin $B_{12}$- and D-tests among Dutch hospital clinicians

The ICCs of the models uncorrected for casemix remained around 9% (ranging from 8.3% to 9.5%) and 9.5% (ranging from 8.5% to 10.1%) for the vitamin $B_{12}$- and

**Table 2** Overview of vitamin B$_{12}$- and D-tests performed among hospital clinicians over the period examined (2015–2019)

| Year | 2015 | 2016 | 2017 | 2018 | 2019 |
|---|---|---|---|---|---|
| No. of hospitals included | 63 | 64 | 66 | 69 | 68 |
| Total no. vitamin B$_{12}$-tests | 275032 | 265546 | 279524 | 279957 | 280522 |
| Total no. vitamin B$_{12}$-tests in hospital with a registered DTC code | 118126 | 163244 | 188063 | 208012 | 213308 |
| Vitamin B$_{12}$-test with a registered DTC without clear indication (%) | 79.0 | 78.0 | 78.2 | 78.3 | 78.1 |
| Total no. of patients with (at least) one vitamin B$_{12}$-test | 233541 | 226999 | 239033 | 240063 | 239351 |
| Total no. of patients with (at least) one vitamin B$_{12}$-test with a registered DTC code | 103540 | 141507 | 162900 | 178933 | 183204 |
| Total no. of patients with (at least) one non-indicated vitamin B$_{12}$-test (with a registered DTC code) | 83549 | 111986 | 128872 | 141914 | 145039 |
| Patients who received at least one vitamin B$_{12}$-test that was considered non-indicated | 80.7 | 79.1 | 79.1 | 79.3 | 79.2 |
| No. of hospitals included | 62 | 61 | 63 | 65 | 65 |
| Total no. vitamin D-tests | 300013 | 300427 | 322703 | 323074 | 336736 |
| Total no. vitamin D-tests in hospital with a registered DTC code | 121892 | 179271 | 208361 | 230620 | 242040 |
| Vitamin D-test with a registered DTC without clear indication (%) | 82.5 | 82.9 | 82.7 | 82.5 | 82.0 |
| Total no. of patients with (at least) one vitamin D-test | 244834 | 247186 | 264811 | 265362 | 272380 |
| Total no. of patients with (at least) one vitamin D-test with a registered DTC code | 101932 | 148660 | 172342 | 189423 | 196452 |
| Total no. of patients with (at least) one non-indicated vitamin D-test (with a registered DTC code) | 84414 | 124162 | 143408 | 157431 | 162345 |
| Patients who received at least one vitamin B$_{12}$-test that was considered non-indicated (%) | 82.8 | 83.5 | 83.2 | 83.1 | 82.6 |

DTC, Diagnosis Treatment Combination.

D-models over time. The ICCs of the casemix-corrected vitamin B$_{12}$- and D-models remained stable around 8.0% (ranging between 7.2% and 8.5%) and 9% (ranging between 8.1% and 9.6%), respectively, throughout the examined period. Online supplemental file 5 contains the ICCs of all models. Casemix correction minimally impacted the calculated ICCs in the case of both the vitamin B$_{12}$- and D-models. The proportion of outpatient visits receiving a non-indicated vitamin B$_{12}$- or D-test over 2019 varied widely among the hospitals included, ranging from 0% to 27.6% for vitamin B$_{12}$ and 0.02% to 34.8% for vitamin D (see figure 1 and online supplemental file 6).

### Correlation in proportions of non-indicated vitamin B$_{12}$- and D-testing

Normality testing (and inspection of density plots) revealed that both the proportions of vitamin B$_{12}$- and D-testing observed among the hospitals are non-normally distributed. The subsequent correlation analysis revealed the presence of a significant positive correlation (r=0.86, p<0.001) between the proportions of non-indicated vitamin B$_{12}$- and D-testing among the included hospitals. Online supplemental file 7 contains the density plots,

normality test results and correlation analysis outcomes for both non-indicated vitamin B$_{12}$- and D-tests.

### DISCUSSION

Between 2015 and 2019, around 78.0% and 82.0% of vitamin B$_{12}$- and D-tests ordered by Dutch hospital clinicians in patients aged 18–70 lacked a clear indication. The total number of vitamin B$_{12}$-tests ordered increased by 2.0%, while the total number of vitamin D-tests increased by 12.2%. Although the total number of vitamin determinations increased, the proportion of patients with at least one vitamin test remained relatively constant (around 80% for vitamin B$_{12}$ and 83% for vitamin D). Women received approximately twice as many vitamin B$_{12}$- and D-tests, as well as non-indicated tests, over the examined period compared with men. Our analysis of hospital variation in non-indicated vitamin B$_{12}$- and D-testing revealed a moderate hospital variation. Furthermore, only a relatively small part of the unexplained variation in non-indicated vitamin B$_{12}$- and D- testing could be ascribed to differences between the hospitals included, suggesting that the problem of non-indicated vitamin testing is

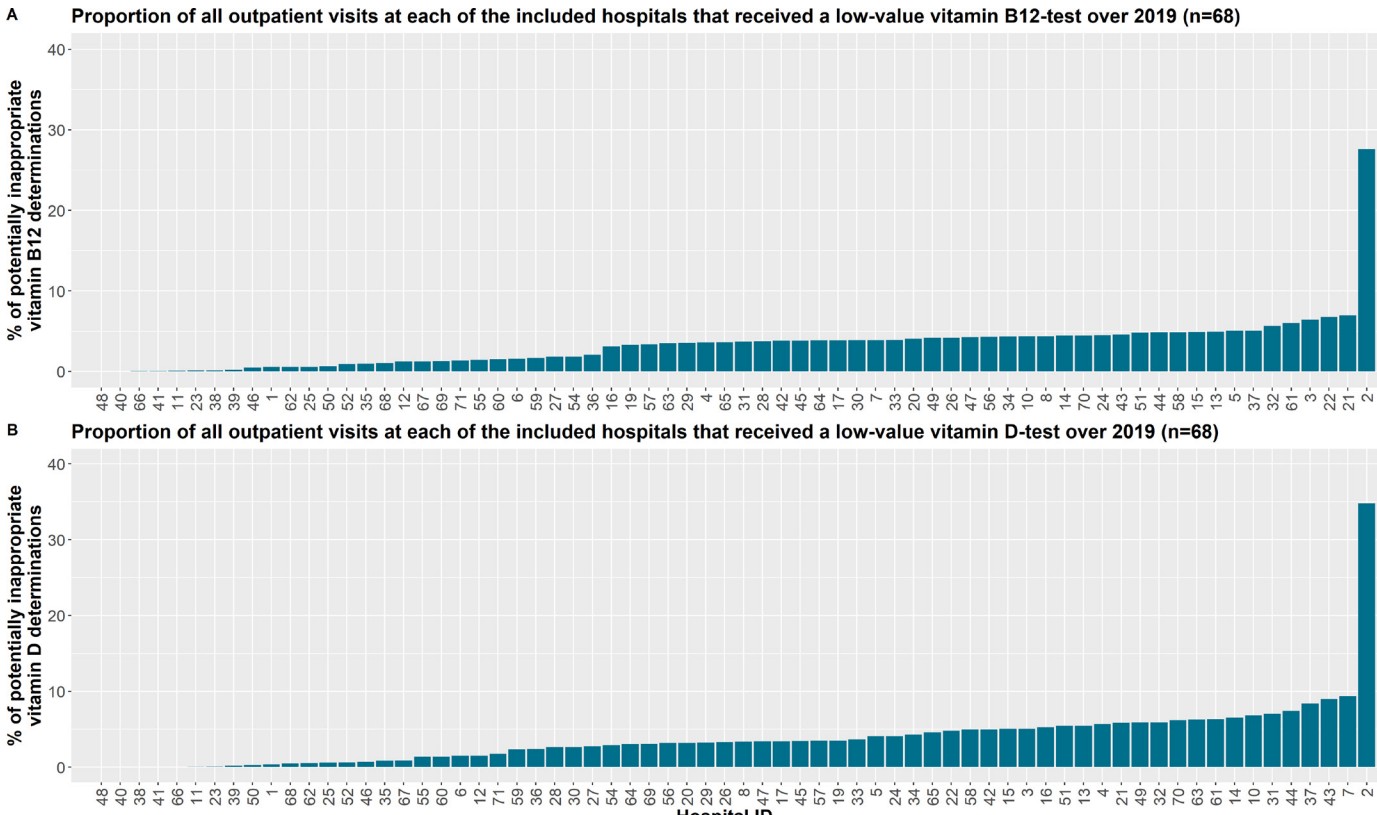

**Figure 1** Proportion of all outpatient visits at each of the included hospitals that received a low-value vitamin (A) B$_{12}$- or (B) D-test over 2019 (n=68).

present among all hospitals. Correlation analysis over 2019 revealed a fairly strong positive (r=0.86) correlation between the rates of non-indicated vitamin B$_{12}$- and D-testing among the hospitals included.

## Comparison to existing literature

The proportion of non-indicated vitamin B$_{12}$- and D-testing among clinicians found in our study (78.0% and 82.0%, respectively) is substantially larger than the proportions reported by other studies. For example, two studies from Italy conducted among hospital clinicians report that on average 17.0% and 6.0% of vitamin B$_{12}$- and D-tests could potentially be non-indicated,[7 18] which is substantially lower than the proportions observed in our study. Our findings are more in line with assessments that are not limited to hospital clinicians. Hence, studies conducted among GPs indicate that between 8.0% and 28.0% of vitamin B$_{12}$-tests[7 15 16 18] and between 7.0% and 91.0% of vitamin D-tests could be considered non-indicated.[7 15 19 21 23] Among these studies, only Naugler *et al* and Gonzalez-Chica reported proportions of non-indicated vitamin D testing which is similar to ours. Naugler *et al*, found that following an intervention the prevalence of non-indicated vitamin D-testing decreased by 91.4%.[19] While Gonzalez-Chica reported 76.5% of vitamin D-tests to be non-indicated after the introduction of new Medicare criteria for rebates. Besides population differences (eg, GPs vs hospital clinicians), varying definitions of non-indicated testing and methods used

could also account for the large differences in assessment outcomes. For example, both our study and that of Naugler *et al* used diagnosis codes to distinguish the appropriateness of vitamin testing.[19] Most assessments of the appropriateness of vitamin testing performed to date used testing intervals or laboratory results to discern non-indicated testing, as did Gonzalez-Chica.[23] This methodological difference might explain the large differences in assessment outcomes. Furthermore, the studies that did not limit their assessment to vitamin testing among GPs were often conducted within a single hospital.[7 18] Moreover, most studies that assessed vitamin B$_{12}$- or D-testing do not specify which type of physicians (GP/hospital clinicians) or vitamin B$_{12}$- and D-tests were included, making comparison to our study challenging.

The absence of clear recommendations regarding the 'appropriate' use of vitamin testing in clinician guidelines does not aid clinicians to appropriately order vitamin testing. The current guidelines for hospital clinicians offer little direction on the appropriate use of vitamin B$_{12}$- or D-tests, leaving clinicians with little guidance when deciding whether to order such tests. These difficulties are magnified by the large knowledge gaps regarding the exact roles of vitamin B$_{12}$ and D within the human body and its metabolism.[34–37] The combination of these factors may provide an explanation for the high proportion of non-indicated testing. Hence, due to the lack of a clear understanding of the roles of vitamin B$_{12}$ and D within the

body and the fear of missing a diagnosis, clinicians may engage in defensive behaviour resulting in the ordering of vitamin tests.

### Strengths and limitations

A strength of this study is its novelty in examining the ordering and appropriateness of vitamin $B_{12}$- and D-testing among hospital clinicians and its variation over several consecutive years. A second strength is that we used a detailed nationally representative database. This allowed us to generate an accurate and reliable overview of the extent of vitamin $B_{12}$- and D-testing among solely hospital clinicians. Furthermore, it also enabled us to accurately distinguish appropriate from inappropriate vitamin testing using diagnosis codes rather than the proxy of testing interval.

However, our study is also prone to limitations. First, we based our distinction of appropriateness mainly on expert opinion and indications derived from the Dutch GP guidelines, as no universal guideline regarding vitamin testing among hospital clinicians exists. We therefore might have misclassified some of the tests from clinicians as being non-indicated. However, we tried to minimise the risk of misclassification by closely collaborating with the involved experts with respect to the creation of the list of indications. Additionally, some differences exist between international and Dutch guidelines regarding indications for vitamin $B_{12}$- and D-testing. For example, guidelines from the USA indicate that vitamin $B_{12}$-testing is considered indicated in patients with cognitive impairments or dementia.[38] However, these are not considered as an indication according to the nationwide guidelines published by the Dutch federation of medical specialists.[10] Potentially, such recommendations could exist in local protocols of Dutch hospitals, which unfortunately are not publicly available. This makes it difficult to compare international assessment outcomes to our study, while subtle differences between guidelines might cause large differences in the used criteria of appropriateness (and subsequently the reported outcomes).

Second, the use of DTC codes enabled us to accurately distinguish the appropriateness of vitamin tests but also has a drawback. DTC diagnosis codes are generally less specific compared with the ICD10 codes described in the used guidelines. We therefore might have misclassified some vitamin tests as being appropriate. Furthermore, although DTC diagnosis codes provide a lot of insight into the diagnosis associated with practices, their registration is prone to misregistration.[39] Clinicians have a vast amount of (often similar) DTC codes to choose from when registering a diagnosis code, thereby adding another layer of complexity to the correct registration of diagnoses. Furthermore, DTC codes are updated as the patient passes through the healthcare system. The registered DTC code therefore does not necessarily represent the initial reason (or diagnosis) for which the vitamin determination was ordered but rather reflects the final diagnosis.

Third, some Dutch hospitals outsource their tests to external commercial laboratories, which are not registered in the LBZ. Our estimate of vitamin testing by clinicians therefore is not complete. However, according to the registration of the National Statistical Office, Statistics Netherlands, 82 hospitals were active in 2019. Since we were able to include data from the majority of the hospitals in the Netherlands in our study (68/82, eg, 83.0% of all hospitals), we do not expect to have missed much in our analysis.[40]

### Implications for research and practice

Our assessment reveals that a large proportion of vitamin $B_{12}$- and D- tests are ordered without a clear indication justifying their use. The total volumes of vitamin $B_{12}$- and D-tests have increased over the years and show no inclination of declining. Based on publicly available fares, we estimate that roughly €3.8 million has been spent on non-indicated vitamin $B_{12}$- and D-tests in 2019 by Dutch hospital clinicians alone.[41] This estimate of the potential savings, however, is very rough, while it only accounts for the cost price of a vitamin $B_{12}$- or D-determination. The observed incidence rates of non-indicated vitamin $B_{12}$- and D-testing, however, suggest that there is ample opportunity to reduce vitamin testing among Dutch clinicians. Especially since non-indicated vitamin $B_{12}$- and D-tests are often ordered for similar diagnoses, and a positive correlation exists between the proportions of non-indicated testing within the same hospitals. We know that there are effective interventions to reduce inappropriate vitamin testing among GPs. A study among Dutch GPs showed that providing both education and feedback successfully reduced the amount of vitamin tests ordered by 20%–25%.[42] Similar interventions might therefore also be effective among hospital clinicians to reduce (non-indicated) vitamin testing. Alternatively, more emphasis could be placed on the institution of fortification and supplementation guidelines to achieve adequate vitamin $B_{12}$ and D intake among the Dutch population. Especially since, the implementation of such guidelines have shown to positively affect vitamin status among the population rendering vitamin testing obsolete in most cases.[43–45] Future research could focus on further examination of patient and physicians' characteristics associated with non-indicated vitamin $B_{12}$- and D-testing. Unfortunately, information regarding the requesting physician (age, sex, etc) of non-indicated tests was not available to us in our study. Insight into physician characteristics associated with non-indicated testing could aid in the design of interventions aiming to address the problem of non-indicated vitamin testing.

### Conclusion

Our research shows that the number of vitamin $B_{12}$-tests slightly increased over the examined time period, while the number of vitamin D tests substantially increased among hospital clinicians. Throughout the examined period, the proportion of $B_{12}$- and D-tests without clear

indication remained high and is substantially higher compared with similar (international) assessments. The observed difference in assessment outcome can potentially be explained by differences in methods and definitions used to identify and define non-indicated vitamin $B_{12}$- and D-tests (eg, the use of associated diagnosis codes instead of test results or proxies such as testing interval). We also observed the presence of moderate hospital variation, but this variation could not be explained by the included patient and hospital characteristics age, sex, SES and hospital size. Hospitals hardly differ in the task they have to undertake: bring down the number of non-indicated $B_{12}$- and D-tests.

**Acknowledgements** The authors would like to thank Professor Dr P. Lips, Professor Emeritus, APH—Aging & Later Life, Professor Emeritus of Internal Medicine, Visiting Fellow Internal Medicine from the Amsterdam UMC, Dr L. de Boer and Dr A. Dittrich for their critical review and contributions to the paper.

**Contributors** Conceptualisation: JLJMM, RBK, SAvD and GPW. Data curation: MZ. Formal analysis: JLJMM and FA. Funding acquisition: RBK and SAvD. Guarantor of overall content: JLJMM. Investigation: JLJMM, RBK, SAvD and HM. Methodology: JLJMM and FA. Project administration: JLJMM. Resources: MZ. Software: JLJMM. Supervision: RBK, SAvD and GPW. Validation: JLJMM, RBK and SAvD. Visualisation: JLJMM. Writing—original draft preparation: JLJMM, RBK and SAvD. Writing—review and editing: JLJMM, RBK, SAvD, MZ, HM, FA and GPW.

**Funding** This work was supported by ZonMw, the Dutch organization for Health Research and Development (grant number 839205002). The funder had no role in the design and conduct of the study.

**Competing interests** None declared.

**Patient and public involvement** Patients and/or the public were not involved in the design, or conduct, or reporting, or dissemination plans of this research.

**Patient consent for publication** Not applicable.

**Ethics approval** The protocol for this research was approved by the Research Ethics Committee of the Radboud University Medical Centre (dossier number 2020-6767). Informed consent of individual patients was not required as anonymised information was obtained from medical records.

**Provenance and peer review** Not commissioned; externally peer reviewed.

**Data availability statement** Data are available upon reasonable request. The source data are not freely available. However, the data used to conduct our analysis are available upon reasonable request after consultation with the data supplier.

**ORCID iDs**
Joris LJM Müskens http://orcid.org/0000-0002-9440-7703
Rudolf Bertijn Kool http://orcid.org/0000-0003-3134-487X
Femke Atsma http://orcid.org/0000-0002-5944-2431
S A van Dulmen http://orcid.org/0000-0003-4003-8540

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
