## [Reviewer comments · BMJ Open]

ARTICLE DETAILS

TITLE (PROVISIONAL)	Non-indicated vitamin B12- and D-testing among Dutch hospital clinicians: a cross-sectional analysis in data-registries
AUTHORS	Müskens, Joris; Kool, Rudolf; Westert, Gert; Zaal, Maarten; Muller, Hein; Atsma, Femke; van Dulmen, SA

VERSION 1 – REVIEW

REVIEWER	Grant, William Sunlight, Nutrition, and Health Research Center I receive funding from Bio-Tech Pharmacal, Inc. (Fayetteville, AR, USA), a supplier of vitamin D supplements.
REVIEW RETURNED	26-Jul-2023

GENERAL COMMENT S	A few things are missing from this manuscript. As my main interest is vitamin D, I will comment on vitamin D. First, there is no information on what the serum 25(OH)D concentrations were. Second, there was only one reference regarding the health benefits of vitamin D and it was from 2007. Third, there is no information regarding guidelines for vitamin D and B12 in The Netherlands. However, there were guidelines for testing, but these guidelines do not seem to be in accordance with emerging science regarding the benefits of vitamin D. Fourth, an alternative to testing would be to institute vitamin D fortification and supplementation guidelines for The Netherlands. See Vitamin D status and current policies to achieve adequate vitamin D intake in the Nordic countries. Itkonen ST, Andersen R, Björk AK, Brugård Konde Å, Eneroth H, Erkkola M, Holvik K, Madar AA, Meyer HE, Tetens I, Torfadóttir JE, Thórisdóttir B, Lamberg-Allardt CJE. Scand J Public Health. 2021 Aug;49(6):616-627. doi: 10.1177/1403494819896878 The positive impact of general vitamin D food fortification policy on vitamin D status in a representative adult Finnish population: evidence from an 11-y follow-up based on standardized 25-hydroxyvitamin D data. Jääskeläinen T, Itkonen ST, Lundqvist A, Erkkola M, Koskela T, Lakkala K, Dowling KG, Hull GL, Kröger H, Karppinen J, Kyllönen E, Härkänen T, Cashman KD, Männistö S, Lamberg-Allardt C. Am J Clin Nutr. 2017 Jun;105(6):1512-1520. doi: 10.3945/ajcn.116.151415. Guidelines for Preventing and Treating Vitamin D Deficiency: A 2023 Update in Poland. Płudowski P, Kos-Kudła B, Walczak M, Fal A, Zozulińska-Ziółkiewicz D, Sieroszewski P, Peregud-Pogorzelski J, Lauterbach R, Targowski T, Lewiński A, Spaczyński R, Wielgoś M, Pinkas J, Jackowska T, Helwich E, Mazur A, Ruchała M, Zygmunt A, Szalecki M, Bossowski A, Czech-Kowalska J, Wójcik M, Pyrzak B, Żmijewski MA, Abramowicz P, Konstantynowicz J, Marcinowska-Suchowierska E,
---

Bleizgys A, Karras SN, Grant WB, Carlberg C, Pilz S, Holick MF, Misiorowski W. *Nutrients*. 2023 Jan 30;15(3):695. doi: 10.3390/nu15030695

34. Holick MF. Vitamin D deficiency. *N Engl J Med*. 2007;357(3):266-81
Comment: This review is good, but more recent reviews should also be cited, e.g. Vitamin D for skeletal and non-skeletal health: What we should know.
Charoenngam N, Shirvani A, Holick MF. *J Clin Orthop Trauma*. 2019 Nov-Dec;10(6):1082-1093. doi: 10.1016/j.jcot.2019.07.004
Immunologic Effects of Vitamin D on Human Health and Disease.
Charoenngam N, Holick MF. *Nutrients*. 2020 Jul 15;12(7):2097. doi: 10.3390/nu12072097.

Vitamin D production rates decrease with increasing age: Vitamin D Synthesis Following a Single Bout of Sun Exposure in Older and Younger Men and Women.
Chalcraft JR, et al., *Nutrients*. 2020 Jul 27;12(8):2237. doi: 10.3390/nu12082237.

“This may help to explain the strong association that has been found between vitamin D deficiency and increased risk for a multitude of diseases, including Alzheimer’s disease, asthma, several autoimmune diseases such as Crohn’s disease, multiple sclerosis, psoriasis, rheumatoid arthritis and ulcerative colitis, many cancers including breast, colon, prostate, sarcomas and skin cancer, chronic pain, dementia, depression, diabetes mellitus, epilepsy, fibromyalgia, falls, fractures and muscle weakness, osteoporosis, osteomalacia, Parkinson’s disease, pregnancy complications including premature birth and death, rickets, schizophrenia and seasonal affective disorder [1,28,35].”

Daily oral dosing of vitamin D3 using 5000 TO 50,000 international units a day in long-term hospitalized patients: Insights from a seven year experience.

McCullough PJ, Lehrer DS, Amend J. *J Steroid Biochem Mol Biol*. 2019 May;189:228-239. doi: 10.1016/j.jsbmb.2018.12.010.

So, who would benefit most from vitamin D supplementation?

Tables 1 & 2. Diagnose description

Diabetes mellitus

Vitamin D and Risk for Type 2 Diabetes in People With Prediabetes : A Systematic Review and Meta-analysis of Individual Participant Data From 3 Randomized Clinical Trials.

Pittas AG, Kawahara T, Jorde R, Dawson-Hughes B, Vickery EM, Angellotti E, Nelson J, Trikalinos TA, Balk EM. *Ann Intern Med*. 2023 Mar;176(3):355-363. doi: 10.7326/M22-3018. Epub 2023 Feb 7.

Pregnancy

Effectiveness of Prenatal Vitamin D Deficiency Screening and Treatment Program: A Stratified Randomized Field Trial.

Rostami M, Tehrani FR, Simbar M, Bidhendi Yarandi R, Minooe S, Hollis BW, Hosseinpanah F. *J Clin Endocrinol Metab*. 2018 Aug 1;103(8):2936-2948. doi: 10.1210/jc.2018-00109.

The Implications of Vitamin D Status During Pregnancy on Mother and her Developing Child.

Wagner CL, Hollis BW. *Front Endocrinol (Lausanne)*. 2018 Aug 31;9:500. doi: 10.3389/fendo.2018.00500. eCollection 2018.

Maternal 25(OH)D concentrations ≥ 40 ng/mL associated with 60% lower preterm birth risk among general obstetrical patients at an urban medical center.

McDonnell SL, Baggerly KA, Baggerly CA, Aliano JL, French CB, Baggerly LL, Ebeling MD, Rittenberg CS, Goodier CG, Mateus Niño JF, Wineland RJ, Newman RB, Hollis BW, Wagner CL. *PLoS One*. 2017 Jul 24;12(7):e0180483. doi: 10.1371/journal.pone.0180483

Hypertension

The Association between Serum 25(OH)D Status and Blood Pressure in Participants of a Community-Based Program Taking Vitamin D Supplements.

Mirhosseini N, Vatanparast H, Kimball SM. *Nutrients*. 2017 Nov 14;9(11):1244. doi: 10.3390/nu9111244.

HIV Effects of sunlight exposure and vitamin D supplementation on HIV patients
NS Akimbekov, RA Ortoski, MS Razzaque - *The Journal of steroid ...*, 2020 - Elsevier

Outcomes not in the tables

Vitamin D food fortification in European countries: the underused potential to prevent cancer deaths.

Niedermaier T, Gredner T, Kuznia S, Schöttker B, Mons U, Lakerveld J, Ahrens W, Brenner H; PEN-Consortium. *Eur J Epidemiol*. 2022 Apr;37(4):309-320. doi: 10.1007/s10654-022-00867-4.

Vitamin D supplementation and total cancer incidence and mortality: a meta-analysis of randomized controlled trials.

Keum N, Lee DH, Greenwood DC, Manson JE, Giovannucci E. *Ann Oncol*. 2019 May 1;30(5):733-743. doi: 10.1093/annonc/mdz059.

An update of the effects of vitamins D and C in critical illness.

Hill A, Starchl C, Dresen E, Stoppe C, Amrein K. *Front Med (Lausanne)*. 2023 Jan 11;9:1083760. doi: 10.3389/fmed.2022.1083760.

Association between vitamin D supplementation and COVID-19 infection and mortality.

Gibbons JB, Norton EC, McCullough JS, Meltzer DO, Lavigne J, Fiedler VC, Gibbons RD. *Sci Rep*. 2022 Nov 12;12(1):19397. doi: 10.1038/s41598-022-24053-4.

Vitamin D is needed before most surgeries

HENRY LAHORE

JUL 26, 2023

https://hlahore.substack.com/p/vitamin-d-is-needed-before-most-surgeries?utm_source=post-email-title&publication_id=1137005&post_id=135448813&isFreemail=false&utm_medium=email

Significant digits. The general rule is that no more non-zero digits should be given than are justified by the uncertainty of the value.

See "Too many digits: the presentation of numerical data"

<https://www.ncbi.nlm.nih.gov/pmc/articles/PMC4483789/>

If the uncertainty (or difference when comparing numbers) is greater than about 7%, only two non-zero digits are justified.

P values should be given to two decimal places unless the first two are 00 or the number lies between 0.045 and 0.054. If the first two are 00, then only one non-zero digit can be given.

Thus, these numbers have too many non-zero digits. Two would probably suffice
Rate of non-indicated vitamin-B12 tests 0.58552

Rho P-value

0.8571538

	Please review all numbers in abstract, text, tables, and figures and adjust accordingly.
--	--

REVIEWER	Crowe, Francesca University of Birmingham, Institute of Applied Health Research
REVIEW RETURNED	18-Aug-2023

GENERAL COMMENTS	The authors have asked an important question about the extent of unjustified testing for levels of blood vitamin B12 and D in patients attending hospitals in The Netherlands. Overall, I find the quality of the statistical analysis and reporting of the results quite poor. The tables are mixed up, the titles of the tables are not very clear, titles of the figures are mixed up and within the tables there are also mistakes. There needs to be a little more attention to detail that is required from the seven authors. The odds ratio estimated from logistic regression approximates the risk ratio (RR) when events are rare; however, when events are common (as in this case), odds ratios from a logistic regression model always overestimate risk ratios (Greenland 1987). Furthermore, the RR is a more intuitive effect estimate than the OR. I would suggest that the authors re-analyse their data to calculate RR using either a log-binomial or robust (or modified) Poisson regression models. However, I see that the authors have not even reported these estimates. Why not? Why did you choose to report the ICC for lack of indication for testing? I am struggling to see what value this is adding. Supplementary files S5 shows proportion of outpatient visits that received a vitamin B12 test without clear indication. Could they also show the results as a proportion of the tests that had a registered DTC code? What is meant to be Table 1 is labelled as Table 3 (page 24). Usually the title of a table should have more descriptive elements such as the years of data collection, where the sample was obtained from etc. Please define the abbreviations below the table. Please apply this to all tables and figures included. Perhaps add in the median and the IQR for the mean number of patients visiting the hospitals as this appears to be skewed (SD is larger than the mean). Table 2 (page 25) which is labelled as Table 4, the last row should read "(%) patients that received at least one vitamin D test that was considered non indicated" For table 3 (on page 25), it would be useful to have some measure of variation. Could the authors please add in the variables that were included in the adjusted model? The title for Figure 1 is confusing. The titles for each figure differs to that to the one below both figures. Minor comments Hein Muller, MD, PhD MD, FACP – what is the affiliation/institution?
---

	Few typos/grammatical errors – would suggest having one more thorough read through to check readability. Page 3, line 22: “diagnose” – should be diagnosis or diagnostic Page 4, line 13: “However, few indications justify the ordering of a vitamin B12- or D-test are described in international guidelines.” – could be “However, there are only a few indications That justify the ordering of a vitamin B12- or D-test which are described in international guidelines.” Page 4, line 18: “Especially, among healthy adults,...” – could be “Among healthy adults,...”
--	--

VERSION 1 – AUTHOR RESPONSE

Reviewer: 1

Mr. William Grant, Sunlight, Nutrition, and Health Research Center

Comments to the Author:

A few things are missing from this manuscript. As my main interest is vitamin D, I will comment on vitamin D.

1. First, there is no information on what the serum 25(OH)D concentrations were.

Response: Even though the reviewer raises a valid point, we have based our distinction of inappropriateness of each of the included B12 and D vitamin determinations on the underlying diagnoses associated with them while information regarding the outcomes of these tests is not recorded within the used database. Furthermore, within the corresponding Dutch guidelines, the decision regarding the appropriateness is rarely coupled to the vitamin serum concentrations but rather disease or disorder related. We are aware of the of the fact that age, gender, food intake and sun exposure all can affect vitamin D levels, and that there are some reports use of vitamin D levels trying use these as markers of several diseases (albeit most often being non-significant or inconclusive). More importantly though, the aim of our study was not examine the correlation between vitamin serum levels and disease, but rather to examine whether hospital clinicians are following the guidelines present for vitamin testing in the Netherlands (in which vitamin concentrations are not regarded as an reason for testing).

2. Second, there was only one reference regarding the health benefits of vitamin D and it was from 2007.

Response: We do agree that the mentioned references regarding both the function of vitamin B12 and D are quite outdated, so added new references of more recent research (line 246).

3. Third, there is no information regarding guidelines for vitamin D and B12 in The Netherlands.

However, there were guidelines for testing, but these guidelines do not seem to be in accordance with emerging science regarding the benefits of vitamin D.

Response: We have examined the amount of low-value vitamin B12- and D-testing among hospital clinicians in the Netherlands. Therefore, we used the guidelines about vitamin B12 and D testing that are applicable and used in the Netherlands. The discussion about the benefits of vitamin B12 and D and supplying these vitamins or not, is very interesting but beyond the scope of this study. We hope that our manuscript aids the discussion of appropriate vitamin testing among hospital clinicians.

4. Fourth, an alternative to testing would be to institute vitamin D fortification and supplementation guidelines for The Netherlands. See Vitamin D status and current policies to achieve adequate vitamin D intake in the Nordic countries.

Response: we agree with the reviewer's comment, that supplementation is a viable alternative to testing. We therefore have added some lines with respect to the supplementation of vitamin B12 and D as being a viable alternative to testing to the discussion section (under the implications for research and practice), lines 301 through 305.

5. Significant digits. The general rule is that no more non-zero digits should be given than are justified by the uncertainty of the value. Please review all numbers in abstract, text, tables, and figures and adjust accordingly.

Response: We have adjusted the amount of digits in the manuscript where applicable (predominantly in the supplementary files).

Reviewer: 2

Dr. Francesca Crowe, University of Birmingham

Comments to the Author:

The authors have asked an important question about the extent of unjustified testing for levels of blood vitamin B12 and D in patients attending hospitals in The Netherlands. Overall, I find the quality of the statistical analysis and reporting of the results quite poor. The tables are mixed up, the titles of the tables are not very clear, titles of the figures are mixed up and within the tables there are also mistakes. There needs to be a little more attention to detail that is required from the seven authors.

6. The odds ratio estimated from logistic regression approximates the risk ratio (RR) when events are rare; however, when events are common (as in this case), odds ratios from a logistic regression model always overestimate risk ratios (Greenland 1987). Furthermore, the RR is a more intuitive effect estimate than the OR. I would suggest that the authors re-analyse their data to calculate RR using either a log-binomial or robust (or modified) Poisson regression models. However, I see that the authors have not even reported these estimates. Why not?

Response: We agree with the notion that the odds ratio of a logistic regression overestimates the risk ratio when events are common (as is the case in our analysis of non-indicated vitamin testing). The rare disease assumption does not hold in this case and odds ratio's should be interpreted as relative odds or, as the reviewer suggests, risk ratio's should be calculated from other types of more advanced models. However, as noted in the method section of our article, the generated models were solely used to assess the intraclass correlation coefficients in order to gain insight into the variation in non-indicated testing between hospitals. The models were not primarily used to gain insight into the risks of covariates on receiving a non-indicated vitamin test. In addition, nowhere in the paper the odds ratio are interpreted as risk ratios, as we are aware of the fact that odds ratio's in our study cannot be interpreted as risk ratios. We therefore also chose not to report ORs in the entire paper to prevent misinterpretation (again, we only were interested in the ICCs). Furthermore, even though the RR could be considered a more intuitive estimate, our choice was still to perform a logistic mixed model and to present odds ratio's together with an overall ICC. [1]

7. Why did you choose to report the ICC for lack of indication for testing? I am struggling to see what value this is adding.

Response: We used the ICC as a measure of the amount of variation in low-value vitamin testing among the hospitals. The intraclass correlation coefficient (ICC) [1, 2], is a metric that can be used when quantitative measurements are made on units that are organized into groups to quantify clustering and variation between clusters. It also enabled us to adjust for case-mix variables in our estimate of this, resulting in a more robust outcome. The ICCs therefore do not refer to the lack of indication among the vitamin tests, but rather reflect the amount of variation in non-indicated testing among the included hospitals. Through this method, we aimed to decipher whether or not non-indicated vitamin testing is limited to several hospitals, or common among all hospitals included.

8. Supplementary files S5 shows proportion of outpatient visits that received a vitamin B12 test without clear indication. Could they also show the results as a proportion of the tests that had a registered DTC code?

Response: the data shown in the table in supplementary file S5 is already limited to the vitamin tests that had an DTC code associated to them. As a result, only the patients who received a determination which had a registered DTC code were included. Furthermore, we chose to report the proportion of outpatient visits receiving a vitamin B12- or D-test without clear indication as opposed to the proportion of tests because this would skew the findings among the included hospitals. Especially since there is a large difference in the amount of patients that visit the different hospitals, we do not think that reporting the raw number of non-indicated or total vitamin B12- or D-tests would give a proper reflection on the matter. We therefore think that it is clearer for readers to only report the proportion of non-indicated testing per outpatient visits.

9. What is meant to be Table 1 is labelled as Table 3 (page 24). Usually the title of a table should have more descriptive elements such as the years of data collection, where the sample was obtained from etc. Please define the abbreviations below the table. Please apply this to all tables and figures included.

Response: we have gone through the manuscript and corrected the references to the tables where necessary. Additionally, we also checked the titles and improved clarity where possible. Abbreviations definitions have been added to figure and table captions where applicable (lines 167 -168).

10. Perhaps add in the median and the IQR for the mean number of patients visiting the hospitals as this appears to be skewed (SD is larger than the mean).

Response: Upon closer inspection of the reported number, we noticed an error was present. We have adjusted this error in table 1 and added the median and IQR for the number of patients visiting the hospital to the manuscript.

11. Table 2 (page 25) which is labelled as Table 4, the last row should read “(%) patients that received at least one vitamin D test that was considered non indicated”

Response: The suggested amendments have been made to the corresponding table.

12. For table 3 (on page 25), it would be useful to have some measure of variation. Could the authors please add in the variables that were included in the adjusted model?

Response: we have added a description of the casemix variables for which the models were corrected to the caption of table 3 (line 472 – 473).

13. The title for Figure 1 is confusing. The titles for each figure differs to that to the one below both figures.

Response: We have adjusted and clarified the title of the figure to match the descriptions as provided in the caption (line 192 – 193).

14. Minor comments:

- Hein Muller, MD, PhD MD, FACP – what is the affiliation/institution?
- Few typos/grammatical errors – would suggest having one more thorough read through to check readability.
- Page 3, line 22: “diagnose” – should be diagnosis or diagnostic
- Page 4, line 13: “However, few indications justify the ordering of a vitamin B12- or D-test are described in international guidelines.” – could be “However, there are only a few indications That justify the ordering of a vitamin B12- or D-test which are described in international guidelines.”
- Page 4, line 18: “Especially, among healthy adults,…” – could be “Among healthy adults,…”

Response: These minor comments all have been addressed and changed in the manuscript.

References

1. Snijders TAB, Bosker RJ. Multilevel Analysis: An Introduction to Basic and Advanced Multilevel Modeling: SAGE Publications; 2011.
2. Koch GG. Intraclass correlation coefficient. Encyclopedia of statistical sciences. 2004.

VERSION 2 – REVIEW

REVIEWER	Crowe, Francesca University of Birmingham, Institute of Applied Health Research
REVIEW RETURNED	02-Feb-2024
GENERAL COMMENTS	I thanks the authors for providing such detailed responses to my comments. all comments have been addressed.

VERSION 2 – AUTHOR RESPONSE